# Obesity Is Associated with Distorted Proteoglycan Expression in Adipose Tissue

**DOI:** 10.3390/ijms24086884

**Published:** 2023-04-07

**Authors:** Astri J. Meen, Atanaska I. Doncheva, Yvonne Böttcher, Simon N. Dankel, Anne Hoffmann, Matthias Blüher, Johan Fernø, Gunnar Mellgren, Adhideb Ghosh, Wenfei Sun, Hua Dong, Falko Noé, Christian Wolfrum, Gunnar Pejler, Knut Tomas Dalen, Svein O. Kolset

**Affiliations:** 1Department of Medical Biology, UIT The Arctic University of Norway, 9019 Tromsø, Norway; 2Department of Nutrition, University of Oslo, 0316 Oslo, Norway; 3EpiGen, Medical Division, Akershus University Hospital, 1474 Nordbyhagen, Norway; 4Department of Endocrinology, Division of Medicine, Akershus University Hospital, 1478 Lørenskog, Norway; 5Mohn Nutrition Research Laboratory, Department of Clinical Science, University of Bergen, 5020 Bergen, Norway; 6Hormone Laboratory, Department of Medical Biochemistry and Pharmacology, Haukeland University Hospital, 5020 Bergen, Norway; 7Helmholtz Institute for Metabolic, Obesity and Vascular Research (HI-MAG) of the Helmholtz Zentrum München at the University of Leipzig and University Hospital Leipzig, 04103 Leipzig, Germany; 8Medical Department III—Endocrinology, Nephrology, Rheumatology, University of Leipzig Medical Center, 04103 Leipzig, Germany; 9Institute of Food, Nutrition and Health, ETH Zurich, 8092 Schwerzenbach, Switzerland; 10Department of Medical Biochemistry and Microbiology, Uppsala University, 75123 Uppsala, Sweden

**Keywords:** extracellular matrix, proteoglycans, obesity, inflammation, adipose tissue

## Abstract

Proteoglycans are central components of the extracellular matrix (ECM) and binding partners for inflammatory chemokines. Morphological differences in the ECM and increased inflammation are prominent features of the white adipose tissues in patients with obesity. The impact of obesity and weight loss on the expression of specific proteoglycans in adipose tissue is not well known. This study aimed to investigate the relationship between adiposity and proteoglycan expression. We analyzed transcriptomic data from two human bariatric surgery cohorts. In addition, RT-qPCR was performed on adipose tissues from female and male mice fed a high-fat diet. Both visceral and subcutaneous adipose tissue depots were analyzed. Adipose mRNA expression of specific proteoglycans, proteoglycan biosynthetic enzymes, proteoglycan partner molecules, and other ECM-related proteins were altered in both human cohorts. We consistently observed more profound alterations in gene expression of ECM targets in the visceral adipose tissues after surgery (among others VCAN (*p* = 0.000309), OGN (*p* = 0.000976), GPC4 (*p* = 0.00525), COL1A1 (*p* = 0.00221)). Further, gene analyses in mice revealed sex differences in these two tissue compartments in obese mice. We suggest that adipose tissue repair is still in progress long after surgery, which may reflect challenges in remodeling increased adipose tissues. This study can provide the basis for more mechanistic studies on the role of proteoglycans in adipose tissues in obesity.

## 1. Introduction

White adipose tissue (WAT) is a dynamic organ that normally expands or shrinks in size according to overall energy status. Expansion of adipose tissue leads to a series of changes that also involves extracellular matrix (ECM) expansion. Notably, adipose tissue expansion may involve a concomitant rise in chronic inflammation, angiogenesis, local hypoxia, fibrosis, adipose tissue dysfunction, and eventually insulin resistance, involving different ECM components [1,2,3].

The ECM is a specialized and dynamic non-cellular network consisting of glycosaminoglycans (GAGs), hyaluronic acid, and different types of collagens and elastic fibers supporting individual cells within the tissues. The expansion of WAT in obesity requires remodeling and reorganization of the ECM, to ensure space for the growth of adipocytes and for the formation of new blood vessels [4,5]. Failure in this process may lead to chronic low-grade inflammation and fibrosis [6]. Many enzymes and molecules are involved in these processes, including matrix metalloproteinases (MMPs) and their inhibitors, i.e., tissue inhibitors of metalloproteinases (TIMPs) [7].

Another crucial component of the ECM is proteoglycans (PGs). This family of glycoconjugates has a wide variety of functions, linked to both inflammation and fibrosis [8]. PGs such as decorin and biglycan bind to and regulate collagens; perlecan and agrin have vital functions in basement membranes; aggrecan and versican are large space-filling PGs in cartilage and vascular tissues; and endocan is a PG with possible links to angiogenesis [9]. Furthermore, the syndecans and glypicans are cell surface PGs with functions related to cell adhesion and ligand interactions, linking the ECM to intracellular signaling [10]. An important feature of PGs is their attached GAG chains, which can bind to and modulate the activities of a wide variety of proteins, including growth factors, chemokines, and protease inhibitors. The GAG chains are highly negatively charged due to the presence of carboxyl and sulfate groups, where sulfation patterns of the GAG chains determine their affinity and specificity for partner interactions [10,11,12]. In addition, certain intracellular PGs, such as serglycin, are secreted after cellular stimuli and have important functions in inflammatory reactions [13]. Furthermore, during the expansion of adipose tissue, resident macrophages will contribute to ECM accumulation [2]. This is achieved through the production of inflammatory chemokines and cytokines, some of which are binding partners of the PGs. In the context of obesity, basement membrane proteins in adipocytes and endothelial cells, such as COL4A1 and HSPG2, were increased in obese patients [14].

To date, there is limited information on the functions of PGs in WAT expansion, inflammation, and fibrosis. Although changes in PGs have been suggested to affect metabolic function and meta-inflammation [12], it is unclear whether PGs play a role in the local and systemic metabolic alterations observed in people with obesity. We have previously demonstrated that the expression of the collagen-binding PG decorin increased in the stromal vascular fraction of adipose tissues when obesity declined after bariatric surgery [15]. Following up on these findings, the main objective of this study was to investigate potential changes in gene expression of ECM components, focusing on PGs in adipose tissues associated with obesity or weight loss.

For this purpose, we focused on the expression levels of PGs, collagens, and PG-binding inflammatory molecules in adipose tissues after weight loss in humans and during adipose tissue expansion in mice.

## 2. Results

### 2.1. Changes in ECM-Related Genes in SC Adipose Tissues following Surgery-Induced Body Weight Loss

To investigate effects on the ECM after a reduction in body weight, we studied alterations in the expression of mRNA encoding ECM components in human SC adipose tissues, before and after bariatric surgery. The first cohort comprised 16 subjects, 12 of which were women (mean age 39.3 ± 10.9 y) with a BMI of 53.3 ± 4.3 kg/m^2^ SD before and 33.1 ± 5.0 kg/m^2^ SD after surgery [16].

The data showed no statistical differences in the expression of the major PGs following body weight loss, except a significant increase in the mRNA expression of PODN/podocan and, as previously reported, a reduced expression of SRGN/serglycin [17]. However, analyses of genes involved in the biosynthesis of the GAG chains of PGs revealed notable differences (Table 1). The mRNA for chondroitin sulfate synthase (CHSY1) responsible for chondroitin sulfate chain elongation, was significantly decreased after surgery, as was the 6-O chondroitin sulfotransferase CHST3. Hyaluronan is one of the main constituents of the ECM, and we found that bariatric surgery resulted in reduced expression of HAS1 and HAS2, two crucial enzymes in hyaluronan synthesis. Moreover, there was also a decrease in the expression of HYAL2, encoding a hyaluronan-degrading enzyme.

Moreover, collagens are essential components of the ECM, and ECM changes in different diseases typically involve members of this protein family. Both TGF-β and fibronectin have profibrotic activities, but the mRNA expression of these molecules was not affected by surgery. Finally, several chemokines and cytokines can interact with the negatively charged GAG chains of the PGs. As shown in Table 2, it is evident that the mRNA expression levels of several such PG-binding chemokines (except for CXCL12) and several interleukins were significantly decreased one year after bariatric surgery. 

To further analyze the implications of PGs for adipose tissue remodeling in different fat depots and whether changes in ECM composition may impact the differential risk profile, we investigated the ECM composition in a second larger cohort, with samples taken from the VIS and SC adipose tissue compartments before and after bariatric surgery that also allowed comparisons of both sexes (Human Cohort 2). The characteristics of these patients are presented in Table 3. SC and VIS adipose tissue samples were obtained and subjected to RNA-seq analyses. 

Targeted analyses of genes encoding PGs and partner molecules revealed that versican (VCAN) was significantly decreased in VIS adipose tissues after surgery, but not in SC adipose tissues (Table 4). Furthermore, syndecan 4 (SDC4) mRNA was significantly increased after surgery in both tissue depots (Table 4). The mRNA expression of glypican 4 (GPC4), a cell surface PG, was significantly increased in VIS adipose tissues after surgery, but not in SC adipose tissues (Table 4). Consistent with the first cohort, PODN mRNA was increased after surgery in SC adipose tissues, although not in the VIS counterpart (Table 4). Osteoglycin is a small keratan sulfate proteoglycan, and our data showed significantly increased OGN mRNA expression in VIS but not in SC adipose tissues after bariatric surgery (Table 4). In contrast, no significant change in OGN expression after bariatric surgery was seen in SC adipose tissues (Table 4). These differences were observed after combining VIS adipose tissues from both males and females. When analyzed separately, the data was the same, except that VCAN mRNA expression was significantly decreased only in VIS adipose tissues from females (not shown). Sex differences in PG expression were therefore not prominent in VIS adipose tissues. 

When comparing the mRNA expression levels of different collagens, COL1A1/collagen I (fibrillar) expression was significantly increased in VIS adipose tissues after bariatric surgery (Table 4), but this was more pronounced in females than in males (not shown). In contrast, COL4A1/collagen IV (basement membrane) mRNA expression was significantly decreased in VIS adipose tissues post-surgery. Finally, analyses were performed to investigate whether the mRNA expression levels of PG-binding chemokines and ECM-degrading enzymes were altered as a result of surgery. Out of a range of analyzed chemokines (CCL2, CCL3, CCL8, CXCL2, IL1B, IL6, and CXCL8), only CCL3 decreased and only in SC adipose tissues in both sexes (result not shown). The mRNA levels of the ECM-degrading enzymes MMP2, MMP9, MMP14, MMP19, MMP25, ADAMTS1, ADAMTS9, and ADAMTS19 were also assessed before and after surgery. Out of these, MMP9 mRNA levels were significantly decreased in both adipose tissue depots after surgery. MMP14 mRNA followed the same trend, but a significant reduction in MMP14 expression was only seen in SC adipose tissues. In comparison, other ECM-degrading enzymes such as MMP19 and MMP25, in addition to ADAMTS1, 9, and 19 were decreased in Human Cohort 1. 

### 2.2. Expression of ECM Components and Pro-Inflammatory Markers in Obese Male and Female Mice

To study the effects of increasing adiposity in a mouse model, we fed female and male mice a low-fat control diet or a high-fat diet for 10 weeks. Both female and male mice had increased total body weights, and SC and VIS adipose tissue organ weights as a result of the high-fat diet intervention (Table 5). 

SC and VIS adipose tissues were recovered and subjected to RT-qPCR analyses to determine the mRNA levels of collagen-binding PGs: lumican (Lum), biglycan (Bcn), decorin (Dcn), as well as Vcan and Sdc4 (Table 6). Bgn expression was significantly increased in SC adipose tissues in both male and female obese mice, but was not altered in VIS adipose tissues. Sdc4 expression was decreased in SC but not VIS adipose tissues of obese compared to lean female mice, whereas males did not exhibit differential expression of Sdc4 in response to the high-fat diet. Lum, Dcn, and Vcam mRNA expression were unaltered in both SC and VIS adipose tissues in obese mice compared to lean controls.

Next, we investigated whether the expression of collagen family members, which comprise the major component of the ECM, was altered after a high-fat diet. mRNA expression for Col1a1/collagen I, Col4a1/collagen IV, and Col6a1/collagen VI were increased in VIS adipose tissues of female mice, but not in the SC counterpart (Table 6). In striking contrast, no significant changes in collagen member expression in response to the high-fat diet were observed in male mice. Both of these mRNAs were increased in expression in VIS adipose tissues from obese female mice, but not in SC adipose tissues of female mice, nor in SC or VIS adipose tissues in males.

Last, we determined the mRNA expression levels for relevant PG-interacting chemokines and cytokines in the two adipose tissue depots. Overall, altered expression of certain pro-inflammatory markers was only observed in VIS adipose tissues. mRNA expression of the chemokines Ccl2 and Cxcl2 was increased in VIS adipose tissues of both female and male obese mice compared to lean controls (Table 6). Moreover, the expression of Ccl8, Il6, and Il1b mRNA was increased in VIS adipose tissues of obese female mice, but not in male mice.

## 3. Discussion

In this study, we demonstrated alterations in the adipose tissue expression of distinct PGs and other ECM-related compounds in human patients after bariatric surgery, as well as after weight gain in mice. The PGs that were affected differed between mice and humans, and also between the two human cohorts analyzed. These data support the concept that alterations in PG expression in obesity are linked to metabolic dysfunction and meta-inflammation [12], and also support the notion of ECM modeling and remodeling as a part of regulatory networks important in regulating adipose tissue homeostasis [18]. In a recent study on ECM changes in human obesity using transcriptomics, the obtained data revealed obesity-associated alterations in the expression of collagens, PGs, and inflammatory mediators [19]. In that study, the expression of three PGs was altered as a result of obesity: serglycin, versican, and asporin.

In our Human Cohort 1, major effects associated with bariatric surgery were seen for genes encoding enzymes responsible for the biosynthesis of the GAG chains of PGs, such as chondroitin sulfate synthase (CHYS3). The same was not observed in Human Cohort 2 (results not shown). However, these two cohorts also differed in the effects of surgery on the PGs affected. The importance of changes in GAG biosynthesis should not be underestimated when PG functions in relation to disease are investigated [20]. Both polymerases, which make GAG chains of the chondroitin sulfate or the heparan sulfate type, epimerases, sulfotransferases, and also enzymes involved in post-translational modifications like the sulfatases, contribute to the construction of biologically efficient and active PGs [21]. Altered expression or enzymatic activity within these different enzyme categories can thus have important consequences for the structure and biological functions of PGs [22]. The effects on GAG enzymes in Human Cohort 1 are important when considering the many and varied biological functions of GAG chains, such as the ability to interact with and modulate the biological activities of chemokines, protease inhibitors, interleukins, and serine proteases [13]. The expression levels of several PG-binding chemokines and interleukins were altered after bariatric surgery or in response to a high-fat diet in mice, suggesting possible coordinated effects on the expression of PGs and PG-binding inflammatory mediators [17].

An intriguing observation was sex differences in gene expression related to bariatric surgery in humans and a high-fat diet in mice. Clearly, such sex-dependent differences in the expression of ECM-related genes in the context of obesity-associated conditions may have biological consequences and be of importance in the design of personalized treatment strategies for obesity-mediated complications. 

The data presented here also highlight biological differences between adipose tissue depots. VIS adipose tissue is generally regarded as more metabolically active, and expansion of this tissue correlates better with the development of insulin resistance than most other adipose tissues [23]. In the data presented here, we consistently observed more profound alterations in gene expression in VIS compared to SC adipose tissues. Based on our study, it will be of interest to evaluate the possible contribution of different ECM components, and their binding partners, to the metabolic regulation in obesity. Changes in ECM turnover associated with loss or gain of body fat, as well as altered expression of enzymes involved in their degradation, suggest a role for ECM turnover in metabolic dysregulation related to obesity. 

Alterations in the ECM represent a natural part of processes such as tissue repair and wound healing, where the ultimate aim is to regain homeostasis after insult or trauma [24]. The expansion of adipose tissues leads to an inflammatory response, followed by tissue repair and potentially adipose tissue fibrosis [4]. Our observation that the expression of some members of the collagen family showed increased expression after bariatric surgery suggests that these processes may be more complicated than previously appreciated. Specifically, our data suggest that adipose tissue repair is still in progress even long after surgery, which may reflect challenges in remodeling massively increased adipose tissues. In line with these findings, we also observed increased collagen expression when mice increase their body fat. These data are consistent with increased adipose tissue fibrosis in obesity, in which COL6A3, encoding one of the major collagens in adipose tissue, may play a causal role in part by regulating inflammatory processes [25,26]. However, it should also be noted that other studies have reported reduced COL6A3 mRNA expression in obesity [16,27], that COL6A3 expression was higher in small compared to large adipocytes [28], and that COL6A3 may prevent adipose tissue inflammation induced by MCP1 [29].

Although the BPD/DS surgery in Human Cohort 1, which is among the most invasive bariatric surgeries performed, induced a major reduction in total nutrient intake and absorption that should be highly consistent across the participants, we cannot rule out an influence on the results of individual differences in pre- or post-surgery diet. In Human Cohort 2, the data and adipose tissue samples were obtained from a biobank and not a trial. Although we acknowledge that dietary intervention for obesity treatment is the basic therapy, we are well aware of difficulties in both adhering to dietary recommendations and monitoring them. We, therefore, added the absence of a structured dietary treatment regimen as the main limitation of our study. However, our adipose tissue donors received frequent and structured healthy diet recommendations that have been used in other trials [30]. On the other hand, we believe that specific dietary programs may have affected the adipose tissue signature in a way that could override any of our cross-sectionally observed relationships with the phenotype.

Both human cohorts originate from different clinical settings and types of surgeries that have induced significant weight loss in patients with obesity. This may have influenced the observed effects on what genes are affected. Indeed, we find differences in mRNA expression of PGs in the two human cohorts and also differences between SC and VIS adipose tissues, which may be due to the different types of surgeries. This inconsistency in regulated genes represents a limitation for the generalization of our results. Since our study focused on mRNA expression levels, further studies combining cell biological approaches and immunohistochemistry are warranted to provide insight into the more functional aspects of ECM turnover in adipose tissue in obesity and during fat loss. In conclusion, our data support and corroborate the concept that the extracellular matrix is crucial for the dynamic adjustment of adipose tissue size [31] and, therefore, in weight regulation.

## 4. Materials and Methods

### 4.1. Human Cohort 1 Adipose Tissues

To study the mRNA expression of PGs and ECM proteins, ECM modifying enzymes, and inflammatory markers in subcutaneous (SC) adipose tissue before and after substantial fat loss, we analyzed previously published data [16]. In this study, subcutaneous biopsies, as well as anthropometric and biochemical data, were recorded from Caucasian Norwegian patients with morbid obesity: 12 women and 4 men (mean age, 39.3 ± 10.9 y SD, and mean BMI, 53.3 ± 4.3 kg/m^2^ SD). The patients underwent the strongly restrictive and malabsorptive bariatric surgery procedure biliopancreatic diversion with duodenal switch (BPD/DS). The patients consumed their habitual diet up until surgery and were at the peak of their obesity. After surgery the patients consumed 200–400 kcal per day during the first week and gradually increased food intake ad libitum, remaining in overall negative energy balance until the 1-year measurement as judged by continued weight loss. The surgery reduced the average BMI from 53.3 +/−4.3 kg/m^2^ pre-surgery to 33.1 +/−5.0 kg/m^2^ 1-year post-surgery. Along with the profound weight loss after surgery, the patients showed strong reductions in systolic blood pressure, glucose, HbA1c, triacylglycerols, insulin, total and LDL cholesterol, and CRP (the latter from a mean ± SD 18.3 ± 12.0 to 3.47 ± 2.61) [16]. Reduced CRP levels after fat loss indicated a reversion of low-grade inflammation. Seven of the patients had type 2 diabetes (five female), taking 1000–2000 mg/day metformin and/or 96–150 IU/day insulin, and in some cases up to 4 antihypertensive drugs, and all medication was ceased after surgery except for one case of low-dose insulin and three diabetes patients on 1-2 antihypertensiva, as previously described [16]. Healthy controls included 13 lean subjects (6 women and 7 men) with BMI < 27 who underwent inguinal hernia repair. The average BMI of the control group was 23 kg/m^2^. The biochemical parameters were normal, except for three CRP samples outside the normal range (6, 8, and 18 mg/L). Total RNA from SC adipose tissues was subjected to microarray analysis using the Illumina iScan system. The patients were not required to follow a specific preoperative diet or change their dietary habits. More information about the study population, approval number, and materials and methods are available in Dankel et al., 2010 [16].

### 4.2. Human Cohort 2 Adipose Tissues

This human cohort comprises paired samples of omental visceral (VIS) adipose tissue and abdominal SC adipose tissue from 55 individuals of the Leipzig Obesity Biobank (LOBB). The biopsies were obtained from individuals with morbid obesity in the context of a two-step bariatric surgery strategy, which in most cases included a sleeve gastrectomy as the first step and laparoscopic Roux-en-Y gastric bypass as the second step. At the first step surgery, patients had obesity grade IV with a BMI > 40 kg/m². Patients with syndromal or monogenetic obesity were not included in this cohort. However, monogenetic obesity was not been formally excluded. All individuals were at either obesity stage 2 or 3 when applying the Edmonton Obesity Staging System [32]. We included patients with obesity onset at an age between 7–10 years (earlier obesity onset has been excluded) or with adult-onset obesity. Patients included in this study lost 49.8 ± 19.8 kg SD for the men and 43.3 ± 17.6 kg SD for the females between the two surgeries. Before the first surgery as well as between first and second step surgeries, patients did not adhere to any specific diet in addition to the individual healthy diet recommendation during regular visits in the obesity management center. Adipose tissue samples were collected during elective laparoscopic abdominal surgery as described previously [33], immediately frozen in liquid nitrogen, and stored at −80 °C. The study was performed in agreement with the Declaration of Helsinki and approved by the Ethics Committee of the University of Leipzig (approval number: 159-12-21052012). All individuals gave written informed consent before participating in the study. Body composition and metabolic parameters were measured as previously described [23]. Study characteristics are given in Table 3. Patients with other known concurrent diseases were not included in the study.

### 4.3. RNA Sequencing of Human Cohort 2

Single-end and rRNA-depleted RNA-seq data were prepared based on the SMARTseq protocol [34]. In brief, RNA was enriched, and reverse transcribed using Oligo(dT) and TSO primers. ISPCR primers were used for cDNA amplification and cDNA was processed with Tn5 using a Nextera DNA Flex kit. All libraries were sequenced on a Novaseq 6000 instrument at the Functional Genomics Center Zurich (FGCZ). Adapter and quality trimming of the raw reads were performed using fastp (v0.20.0) [35] considering a minimum read length of 18 nts and a quality cut-off of 20. Sequence pseudo alignment of the resulting high-quality reads against the human reference genome (build GRCh38.p13) and gene level expression quantification (gene model definition from GENCODE release 32) was carried out using Kallisto (Version 0.46) [36]. Samples with more than 20 million mapped read counts were downsampled to 20 million read counts using the subsampleCountMatrix function of the R package ezRun v3.14.1 (https://github.com/uzh/ezRun, accessed on 23 March 2022). Raw count data were homoscedastically normalized with respect to library size using the variance stabilizing transformation from DESeq2 (v1.32.0) [37]. Since we have paired data for SC and VIS adipose tissues, and the patient ID represented a large source of variance, an adjustment was performed to account for differences between the patients. Further, gene expression counts were calibrated with transcript integrity numbers (TINs) which could effectively neutralize in vitro RNA degradation effects by reducing false positives and recovering biologically meaningful pathways [38]. The TIN was calculated for each sample by running the Tin.py script from the RSeQC package v4.0.0 [38] using Snakemake workflow v6.4.1 [39].

### 4.4. Animal Experiment

Animal experiments conformed to the ARRIVE guidelines and ethical guidelines in the European Directive 2010/63/EU on the protection of animals used for research purposes. The experimental protocols were approved by the Norwegian Animal Research Authority (Mattilsynet, approvals FOTS #10901). The mice were housed in individual ventilated cages (IVC) at 22–24 °C with a strict 12 h light/dark cycle.

Male and female mice with a C57BL/6N genetic background (Janvier Labs, Le Genest Saint Isle, France) were bred and housed at the animal facility of the University of Oslo as described previously [17]. In short, age-matched male and female mice were housed with ad libitum access to a diet with a low-fat content (cat #D12450J, Research Diets, New Brunswick, NJ) or a diet with a high-fat content (cat # D12492, Research Diets) from 8 to 18 weeks of age. The low-fat diet contained 70 E% from carbohydrates (corn starch, sucrose, and maltodextrin), 10 E% from fat (soybean oil and lard), and 20 E% from protein (casein). The high-fat diet contained 20 E% from carbohydrates (corn starch, sucrose, and maltodextrin), 60 E% from fat (soybean oil and lard), and 20 E% from protein (casein). At the end of the diet intervention, all mice were sacrificed by cervical dislocation. Gonadal VIS and intraperitoneal SC adipose tissues were rapidly frozen in liquid nitrogen. Body and organ weights are listed in Table 5.

### 4.5. RNA Isolation and Reverse Transcription Quantitative PCR (RT-qPCR) of Mouse Adipose Tissue

Adipose tissues (n = 8) were homogenized in Qiazol (Qiagen, Hilden, Germany) using a Precellys 24 homogenizer (Bertin Technologies, Montigny-le-Bretonneux, France). Homogenized tissues were then centrifuged at 10,000× *g*, at 4 °C for 15 min to remove lipids and cell debris. The cleared homogenate was extracted with chloroform and centrifuged at 14,000× *g*, at 4 °C for 15 min. The upper phase was mixed with alcohol and loaded onto Nucleospin RNA purification columns (Macherey-Nagel, Düren, Germany) to purify total RNA. Total RNA was reverse transcribed into cDNA using Multiscribe^®^ RT kit (Thermo Fisher, Waltham, MA, USA). RT-qPCR was performed with the Bio-Rad SsoAdvanced™ Universal SYBR^®^ Green Supermix using 5–10 ng/µL cDNA. The 60S ribosomal protein L32 (Rpl32) was verified as stably expressed in VIS and SC adipose tissues and used to normalize the gene expression data. Relative gene expression was calculated pairwise for every tissue depot (lean vs. obese) by the relative quantification method (2−ΔΔCq).

### 4.6. Statistical Analyses

Human Cohort 2: Between-subject comparisons for specific genes were performed with a nonparametric statistical approach using the R package ggstatsplot (v0.9.1) based on one-way Kruskal–Wallis ANOVA and pairwise Dunn’s tests [40]. The Shapiro–Wilk test was used to check the normality distribution of the data. *p*-values were corrected for multiple inference using the Holm method. Analyses were performed using R version 4.1. Only significant adj. *p*-values for pre- and post-surgery within each adipose tissue are reported.

Animal experiments: RT-qPCR data are presented as mean ± 95% CI. Significant differences in the groups were estimated with two-way ANOVA by taking the diet intervention and the two adipose tissue depots as variables (* *p* < 0.05; ** *p* < 0.01; *** *p* < 0.001, **** *p* < 0.0001). The Shapiro–Wilk test was used to test the normality of the data. The obtained data were visualized in GraphPad Prism version 6.04 (La Jolla, CA, USA).

## Figures and Tables

**Table 1 ijms-24-06884-t001:** GAG enzymes.

Gene Symbol	Gene ID	Definition	Fold Change	D[i]	De[i]	Delta[i]	FSN[i]	FDR[i]	q-val[i]
CHSY1	22856	Chondroitin sulfate synthase 1	−2.04	−7.83	−2.013	5.824	0	0	0
HAS2	3037	Hyaluronan synthase 2	−2.16	−6.327	−1.798	4.529	0	0	0
CHST12	55501	Carbohydrate (chondroitin 4) sulfotransferase 12	1.19	5.885	1.83	4.055	0	0	0
XYLT2	64132	Xylosyltransferase II	1.33	5.743	1.79	3.952	0	0	0
CHST3	9469	Carbohydrate (chondroitin 6) sulfotransferase 3	−1.66	−5.345	−1.64	3.706	0	0	0
HAS1	3036	Hyaluronan synthase 1	−2.80	−5.225	−1.615	3.611	0	0	0
HYAL2	8692	Hyaluronoglucosaminidase 2	−1.51	−5.153	−1.602	3.551	0	0	0
HAPLN3	145864	Hyaluronan and proteoglycan link protein 3	−1.27	−4.742	−1.535	3.207	0	0	0
CHST14	113189	Carbohydrate (N-acetylgalactosamine 4-0) sulfotransferase 14	1.18	4.404	1.399	3.005	0	0	0
XYLT1	64131	Xylosyltransferase I	−1.07	−2.454	−1.076	1.378	42.923	0.728	0.727

D[i], observed difference; De[i], expected difference; Delta[i], difference between observed and expected difference (D[i]- De[i]); FSN[i], false significant numbers (number of called significant genes expected to be false); FDR[i], false discovery rate (100*FSN[i]/called); q-val[i], lowest FDR.

**Table 2 ijms-24-06884-t002:** Chemokines and interleukins.

Gene Symbol	Gene ID	Definition	Fold Change	D[i]	De[i]	Delta[i]	FSN[i]	FDR[i]	q-val[i]
CXCL2	2920	Chemokine (C-X-C motif) ligand 2	−8.36	−13.35	−2.713	10.637	0	0	0
CX3CL1	6376	Chemokine (C-X3-C motif) ligand 1	−2.91	−13.217	−2.686	10.531	0	0	0
CCL2	6347	Chemokine (C-C motif) ligand 2	−8.07	−9.914	−2.288	7.626	0	0	0
CCL8	6355	Chemokine (C-C motif) ligand 8	−4.41	−9.118	−2.19	6.928	0	0	0
CCL3	6348	Chemokine (C-C motif) ligand 3	−2.29	−5.683	−1.697	3.986	0	0	0
CCL3L3	414062	Chemokine (C-C motif) ligand 3-like 3	−2.78	−5.323	−1.634	3.69	0	0	0
CXCL12	6387	Chemokine (C-X-C motif) ligand 12	1.54	5.228	1.628	3.6	0	0	0
CCL3L1	6349	Chemokine (C-C motif) ligand 3-like 1	−1.75	−4.739	−1.532	3.207	0	0	0
CCL4L1	9560	Chemokine (C-C motif) ligand 4-like 1	−2.07	−4.057	−1.401	2.656	0	0	0
IL8	3576	Interleukin 8 (IL8), mRNA.	−19.53	−14.895	−3.001	11.894	0	0	0
IL1B	3553	Interleukin 1, beta (IL1B), mRNA.	−4.76	−11.155	−2.449	8.706	0	0	0

D[i], observed difference; De[i], expected difference; Delta[i], difference between observed and expected difference (D[i]- De[i]); FSN[i], false significant numbers (number of called significant genes expected to be false); FDR[i], false discovery rate (100*FSN[i]/called); q-val[i], lowest FDR.

**Table 3 ijms-24-06884-t003:** Cohort 2 patient characteristics.

	Male		Female	
	Surgery Step 1	Surgery Step 2	Surgery Step 1	Surgery Step 2
Age (years)	45.1 ± 7.5	48.0 ± 8.5	42.24 ± 9.9	45.7 ± 10.8
Height (m)	1.82 ± 0.08	1.82 ± 0.08	1.67 ± 0.07	1.67 ± 0.07
WHR	1.06 ± 0.08	1 ± 0.05	1.01 ± 0.11	1 ± 0.13
Waist (cm)	150 ± 12.42	119.11 ± 12.63	145.28 ± 15.71	117.5 ± 18.86
Weight (kg)	181.49 ± 22.95	131.66 ± 20.23	155.41 ± 27.07	112.11 ± 22.56
BMI (kg/m^2^)	55.5 ± 9.1	40.2 ± 6.4	55.8 ± 9.5	40.4 ± 7.7
Weight loss (kg)	-	49.8 ± 19.8	-	43.3 ± 17.6
ASAT (µkat/L)	0.73 ± 0.4	0.52 ± 0.33	0.71 ± 0.4	0.36 ± 0.16
ALAT (µkat/L)	0.52 ± 0.12	0.52 ± 0.22	0.61 ± 0.36	0.35 ± 0.11
FPG (mmol/L)	7.32 ± 4.88	5.53 ± 1.48	6.11 ± 2.13	5.55 ± 1.42
FPI (pmol/L)	157.58 ± 134.23	66.26 ± 47	206.65 ± 206.3	104.3 ± 68.14
Concomitant diseases:				
T2D (N)	18	8	30	10
Arterial hypertension	14	5	11	4
Obstructive sleep apnea	4	0	2	1
Dyslipidemia	5	5	4	4
Chronic joint pain	8	4	7	5
Concomitant medications:				
Metformin	14	5	25	7
GLP-1 receptor agonist	2	0	1	0
Insulin	3	1	4	0
RAS inhibitors	14	5	11	4
Betablockers	8	7	7	6
Diuretics	7	5	4	2
Statins	5	5	4	4

WHR, waist–hip ratio; BMI, body mass index; FGP, fasting plasma glucose; FPI, fasting plasma insulin; GLP-1, glucagon-like peptide-1; RAS, renin–angiotensin system; T2D, type 2 diabetes. Values are given as mean ± SD.

**Table 4 ijms-24-06884-t004:** *p*-values of versican (VCN), syndecan 4 (SDC4), glypican 4 (GPC4), podocan (PODN), and osteoglycan (OGN) in subcutaneous (SC) and visceral (VIS) adipose tissues pre- and post-surgery in Human Cohort 2. mRNA expression is variance stabilizing transformation (VST) normalized. A Kruskal–Wallis one-way ANOVA was applied and for pairwise comparisons, Dunn’s test was used and corrected for multiple inferences using the Hommel method.

	Gene Expression Pre- vs. Post-Surgery	
	Subcutaneous Adipose Tissue	Visceral Adipose Tissue
VCAN	0.34	0.000309 ***
SDC4	0.049 *	0.00602 **
GPC4	0.16	0.00525 **
PODN	0.00633 **	0.41
OGN	0.63	0.000976 ***
COL1A1	0.72	0.00221 **
COL4A1	0.09	0.03 *
CCL3	0.0000328 ***	0.08
MMP9	0.000000141 ***	0.000256 ***
MMP14	0.02 *	1.00

* represents *p* < 0.05, ** *p* < 0.01 and *** *p* < 0.001.

**Table 5 ijms-24-06884-t005:** Body weights and organ weights of lean and obese C57BL/6N mice (18 weeks old).

	Male		Female	
	Lean	Obese	Lean	Obese
Body weight (g)	33.5 ± 3.2	43.8 ± 5.1 ****	26.2 ± 2.2	31.2 ± 6.1 *
Heart (g)	0.138 ± 0.02	0.130 ± 0.007	0.102 ± 0.006	0.097 ± 0.007
Liver (g)	1.55 ± 0.17	1.68 ± 0.21	1.18 ± 0.14	1.02 ± 0.13
Visceral AT (g)	0.82 ± 0.48	2.42 ± 0.46 ****	0.73 ± 0.23	1.89 ± 0.83 ****
Subcutaneous AT (g)	0.48 ± 0.27	1.42 ± 0.47 ****	0.54 ± 0.20	1.15 ± 0.60 **
Kidneys (g)	0.332 ± 0.05	0.377 ± 0.04	0.246 ± 0.02	0.240 ± 0.02

AT, adipose tissue. Data are presented as mean ± SD, n = 8–15. * *p* < 0.05, ** *p* < 0.01, **** *p* < 0.0001 indicates significant difference between lean and obese mice.

**Table 6 ijms-24-06884-t006:** Relative mRNA expression of proteoglycan, collagen, chemokines, and interleukins in intraperitoneal subcutaneous (SC) and visceral (VIS) adipose tissues of obese mice fed a high-fat diet for 10 weeks compared to those in lean mice fed a low-fat diet. A two-way ANOVA test was used where obesity and adipose tissue depots were taken as variables. The Shapiro–Wilk test was used to test the normality of the data.

		Male Lean vs. Obese				Female Lean vs. Obese			
		SC		VIS		SC		VIS	
Gene Symbol	Definition	FC	*p*-Value	FC	*p*-Value	FC	*p*-Value	FC	*p*-Value
Lum	Lumican	0.97	0.995	1.61	0.101	1.24	0.998	2.23	0.136
Bgn	Biglycan	2.21	0.008	1.71	0.200	6.37	0.001	2.53	0.454
Dcn	Decorin	0.67	0.241	0.58	0.110	1.14	0.954	2.04	0.086
Vcan	Versican	0.80	0.591	1.31	0.139	0.77	0.396	1.13	0.761
Sdc4	Syndecan 4	1.26	0.089	1.20	0.212	0.71	0.027	0.97	0.991
Has1	Hyaluronan synthase 1	0.46	0.186	0.97	0.961	2.05	0.356	3.17	0.022
Col1a1	Collagen type I alpha 1 chain	1.05	0.980	1.09	0.945	2.18	0.171	2.76	0.028
Col4a1	Collagen type IV alpha 1 chain	1.25	0.668	0.81	0.774	2.37	0.160	3.85	0.0012
Col6a1	Collagen type VI alpha 1 chain	1.85	0.184	0.94	0.987	2.31	0.158	3.78	0.001
Tgfb	Transforming growth factor beta 1	1.12	0.950	1.42	0.538	1.49	0.937	3.29	0.0013
Ccl2	Chemokine (C-C motif) ligand 2	2.73	0.196	8.29	0.0001	3.19	0.271	10.69	0.0001
Ccl3	Chemokine (C-C motif) ligand 3	0.44	0.223	1.00	0.99	0.48	0.40	0.49	0.41
Ccl8	Chemokine (C-C motif) ligand 8	0.38	0.277	1.37	0.673	0.78	0.965	4.37	0.003
Cxcl2	Chemokine (C-X-C motif) ligand 2	2.52	0.329	4.10	0.018	2.76	0.288	7.66	0.0001
Il6	Interleukin 6	0.71	0.629	1.28	0.676	1.17	0.871	2.45	0.006
Il1b	Interleukin 1 beta	0.71	0.639	1.39	0.466	0.96	0.999	3.39	0.012

## Data Availability

Additional data are available on request from the corresponding author.

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
