# Peer review of "Obesity Is Associated with Distorted Proteoglycan Expression in Adipose Tissue"

_ijms, 2023, doi:10.3390/ijms24086884_

Round 1
Reviewer 1 Report
While the paper addresses an interesting issue, its main contribution is unclear. In particular,
- the originality of the paper needs to be worked out before it can be considered for publication;
- the authors discussed the effects of bariatric surgery on gene expression of ECM-related compounds, without properly examining the effects of obesity on the same pathways (is the mRNA expression of PG and collagen normalized or not?) as they did for mice. As a consequence, it is not always easy to follow the logic;
- the experimental analysis are too underdeveloped and, as the authors stated in the last part of the discussion, further analyses (e.g. immunohistochemistry) are necessary.
Reviewer 2 Report
Comments to the Authors of manuscript number: ijms-2249843 entitled “Obesity is associated with distorted proteoglycan expression in adipose tissue”.
The study includes two parts. One was performed on human tissue and the second on mice. It is very interesting, however it should be corrected.
1. L 39- or WAT?
2. L 51- are
3. L 55 – the reference should be added
4. Results are not discussion, thus this part should be rephrased.
5. Discussion cannot present the same information that are presented in the part of results.
6. L 415 – how results were normalized?
Reviewer 3 Report
The manuscript entitled „Obesity is associated with distorted proteoglycan expression in adipose tissue” presents interesting issue, but some problems should be corrected.
Abstract:
A number of redundant information are included while the necessary ones are not presented. Authors should follow instructions for authors for the content of this section, and should provide all necessary information (it is even more important than word count).
The aim of the study should be formulated.
Authors should present the methodology of the study (studies).
The specific numeric results should be presented and accompanied by the results of the statistical analysis (p-Value).
Introduction:
Authors should not focus on presenting their own previous studies, but the general global state of knowledge.
Authors should not present methodology and results of the study in this section (lines 80-85)
The aim of the study should be formulated.
Results:
The detailed characteristics of the studied human subjects should be presented – including type of obesity, concurrent illnesses, applied diet, etc, as without it is hard to interpret the presented results.
Instead of figures Authors should rather present tables, as their figures are extremely hard to follow
Discussion:
Instead of reproducing previously presented results, Authors should rather discuss them based on the literature.
The limitations of the study should be presented and discussed.
Materials and Methods:
Authors should describe how did they control the diet of human subjects (I suppose that they did it) and what was the diet of animals (composition and nutritional value).
Authors should describe the verification of the distribution of data
Authors Contributions:
Based on the information presented it seems that some authors did not participate in preparing the manuscript, so they should not be presented as authors of the study but rather in the Acknowledgements Section. There is a serious risk of guest authorship procedure that is forbidden.
Round 2
Reviewer 3 Report
The manuscript entitled „Obesity is associated with distorted proteoglycan expression in adipose tissue” presents interesting issue, but some problems should be corrected.
Major:
The problem is associated with the fact that during the study Authors did not control the diet of their participants – it is a serious limitation of the study
Abstract:
The specific numeric results should be presented and accompanied by the results of the statistical analysis (p-Value).
Introduction:
The aim of the study should be formulated.
Results:
The detailed characteristics of the studied human subjects should be presented – including type of obesity, concurrent illnesses, applied diet, etc, as without it is hard to interpret the presented results.
Instead of figures Authors should rather present tables, as their figures are extremely hard to follow
Discussion:
Authors should discuss them based on the literature – more discussion is needed in this Discussion Section
All the limitations of the study should be presented and discussed (there are numerous limitations that were not included)
Authors Contributions:
Authors should be consistent – who is AID? There is no such Author of their study
Round 3
Reviewer 3 Report
The detailed characteristics of the studied human subjects should be presented – including type of obesity, concurrent illnesses, applied diet, etc, as without it is hard to interpret the presented results.
